# A Theory-Informed Systematic Review of Barriers and Enablers to Implementing Multi-Drug Pharmacogenomic Testing

**DOI:** 10.3390/jpm12111821

**Published:** 2022-11-02

**Authors:** Essra Youssef, Debi Bhattacharya, Ravi Sharma, David J. Wright

**Affiliations:** 1School of Pharmacy, University of East Anglia, Norwich Research Park, Norwich NR4 7TJ, UK; 2School of Healthcare, University of Leicester, University Road, Leicester LE1 7RH, UK; 3Bedfordshire Hospitals, NHS Foundation Trust, Kempston Road, Bedford MK42 9DJ, UK; 4Centre fpr Pharmacy, University of Bergen, 5007 Bergen, Norway

**Keywords:** pharmacogenomics, implementation, systematic review

## Abstract

PGx testing requires a complex set of activities undertaken by practitioners and patients, resulting in varying implementation success. This systematic review aimed (PROSPERO: CRD42019150940) to identify barriers and enablers to practitioners and patients implementing pharmacogenomic testing. We followed PRISMA guidelines to conduct and report this review. Medline, EMBASE, CINAHL, PsycINFO, and PubMed Central were systematically searched from inception to June 2022. The theoretical domain framework (TDF) guided the organisation and reporting of barriers or enablers relating to pharmacogenomic testing activities. From the twenty-five eligible reports, eleven activities were described relating to four implementation stages: ordering, facilitating, interpreting, and applying pharmacogenomic testing. Four themes were identified across the implementation stages: IT infrastructure, effort, rewards, and unknown territory. Barriers were most consistently mapped to TDF domains: memory, attention and decision-making processes, environmental context and resources, and belief about consequences.

## 1. Introduction

The field of pharmacogenomics—testing genetic material to inform prescribing choices—is complex and rapidly evolving. Despite early enthusiasm [1,2], glob implementation of pharmacogenomic (PGx) testing into routine clinical care has been uneven [3].

The recent expansion of pharmacogenomic clinical guidelines and recommendations [4]; efforts to harmonise testing and reporting practices [5,6,7,8,9] and reduced testing costs have driven the increase in the number of institutions adopting PGx testing in the last ten years [10]. However, challenges to widespread implementation remain and include among the most reported, factors such as gaps in knowledge, data storage and security, drug–gene pair selection, and legal and ethical concerns [6].

Within healthcare, implementing new practices such as PGx testing requires changes in the behaviour of relevant actors. Implementation science can help reveal the determinants of healthcare professionals’ current and desired behaviours related to PGx testing [11]. Thus far, implementation of PGx testing in clinical practice has been explored through expert reviews [3,6] that describe barriers and enablers broadly at a macro level, without the application of implementation theory. The absence of theory potentially slows the collective understanding of PGx testing implementation challenges as researchers lack insight into the structural and psychological processes regulating the behaviours observed. This, therefore, limits the ability to design evidence-based solutions for the implementation problems observed and instead creates a reliance on intuition and personal knowledge.

The theoretical domain framework (TDF) is an integrative framework which synthesises 33 theories and 128 constructs of behaviour and behavioural change [12,13]. It comprises 14 domains representing determinants of behaviour; for example, the ‘knowledge’ domain refers to an ‘awareness of the existence of something’ and ‘memory, attention and decision making’ refers to ‘the ability to retain information, focus selectively on aspects of the environment, and choose between two or more alternatives’ [13]. The 14 domains are linked to a taxonomy of behavioural change techniques (BCTs) [14], which ate the “building blocks” of behavioural change interventions. Compared to other implementation frameworks such as the Consolidated Framework of Implementation Research (CFIR), the TDF provides a comprehensive lens to understand healthcare provider behaviours, versus the CFIR which is an “over-arching typology” for understanding implementation [15]. Another advantage of the TDF is it is linked to evidence-based behavioural change techniques [16]. In this way, it can be used to ‘diagnose’ the implementation problem and equip the researcher with the tools to overcome the problem. The TDF has been used to identify factors influencing the behaviours of nurses and physicians related to implementing genomic medicine in secondary and tertiary care settings [17].

We proposed using the TDF to organise the published literature on barriers and enablers to the implementation of pharmacogenomic testing. To achieve this, we undertook a systematic review of the published literature to identify and synthesise all barriers and enablers to behaviours of prescribers, pharmacists, and patients related to implementing multi-drug pharmacogenomic testing. We then mapped these barriers and enablers to domains within the TDF, with the purpose of identifying the most appropriate configuration for PGx testing implementation.

## 2. Materials and Methods

This systematic review was registered with PROSPERO (CRD42019150940) and conducted in accordance with the Preferred Reporting Items for Systematic reviews and Meta-Analyses (PRISMA) statement [18]. PubMed, Cochrane Database of Systematic Reviews, PROSPERO, and the Joanna Briggs Institute Systematic Reviews database were searched to ensure this systematic review would not duplicate existing work.

### 2.1. Search Strategy

The search strategy was developed through a scoping search and in consultation with an information services librarian. MEDLINE (via OVID); EMBASE (via OVID); CINAHL (via EBSCO); PsycINFO (via EBSCO) and PubMed Central were searched on 25 November 2019 with no restrictions. Further articles were elicited by backward searching lists of included articles. We updated the search on 1 January 2022 and 10 June 2022. The detailed search strategy is available in Appendix A.

### 2.2. Inclusion and Exclusion Criteria

Inclusion and exclusion criteria were developed using the PICO framework [19]. Articles were included if they were published in English in a peer-reviewed journal and reported real-world experiences of implementation of multi-drug pharmacogenomic testing. Articles were excluded if they reported on single-drug pharmacogenomic testing, somatic gene testing, whole genome sequencing, or paediatric pharmacogenomic testing. Systematic reviews and articles without a qualitative component were excluded. No date or country restrictions were enforced.

### 2.3. Screening and Extraction

All retrieved articles were imported into EndNote Library X9, and duplicates were removed. Two reviewers (E.Y. and R.S.) independently screened all articles against the inclusion and exclusion criteria by title and abstract and then by full text. Interrater concordance was calculated using the Cohens Kappa statistic [20]. Disagreements were resolved through discussion. A data extraction tool was developed in Microsoft Excel. Data extraction and mapping were undertaken independently by one reviewer (E.Y.) and checked by another reviewer with expertise in behavioural change theory and use of the TDF (D.B.) Disagreements were resolved through discussion.

### 2.4. Data Synthesis Process

We used framework analysis with the TDF as an a priori framework to map determinants to the behaviours of prescribers, pharmacists, and patients related to the implementation of PGx testing. Table 1 shows descriptions of each of the TDF domains in the context of PGx test implementation.

Figure 1 shows the data synthesis process. Extracted data items were first grouped into behavioural descriptions. If the data item did not adequately correspond to an existing behavioural description, a new behavioural description was created. Next, data items related to barriers or enablers to each behavioural description were extracted and mapped to domains in the TDF. We then coded whether these determinants were reported by prescribers, pharmacists, or patients both directly or indirectly through author interpretations. Finally, we grouped the determinants into overarching themes [21] and described the findings narratively.

### 2.5. Quality of Reporting Assessment

A range of quality assessment tools (Centre for Evidence-Based Management [22]; The Critical Appraisal Skills Programme CASP Qualitative checklist [23]; and The Critical Appraisal of Survey [24]) were used according to the study designs [25].

## 3. Results

Figure 2 provides the flow of studies from 1515 studies retrieved to the 27 included studies. The primary reasons for exclusion at full-text screening were the implementation of single drug/gene pair testing rather than multi-drug or being unable to isolate the reported barriers and enablers to pharmacogenomic implementation from those for wider genomic implementation. Interrater concordance according to Cohens Kappa was calculated as 0.87, 0.72, and 0.78 at the screening titles, abstracts, and full-text stages, respectively.

### 3.1. Characteristics of Studies

Table 2 summarises the characteristics of the twenty-seven included articles all of which were from high-income countries and primarily the United States (n = 23, 85%). Both primary and secondary care settings were represented. Seventeen articles explored behaviours of prescribers [26,27,28,29,30,31,32,33,34,35,36,37,38,39,40,41,42,43], fifteen explored behaviours of the pharmacist [28,30,32,33,41,42,43,44,45,46,47,48,49,50,51] and seven explored patient behaviours [27,36,39,45,48,51,52]. Most of the included studies collected data via document analysis and surveys. There were no differences in reported barriers and enablers between different study designs.

### 3.2. Target Behaviour Areas

Four implementation stages were described across the reports. These stages are shown in Table 3. Each implementation stage incorporated multiple behaviours of which the ‘facilitating test’ stage comprised the most activities.

### 3.3. Themes

Figure 3 illustrates the four themes that emerged from the data with corresponding TDF domains. These themes covered all stages of the implementation cycle and were IT Infrastructure, Effort, Rewards, and Unknown territory.

#### 3.3.1. IT Infrastructure

All implementation stages had barriers and enablers related to the extent to which local information technology systems were adapted. The majority of the barriers and enablers within this theme were mapped to two TDF domains: ‘Memory, attention and decision making′ and ‘Environmental context and resources’.

Several papers reported ways in which technology was or could be utilised to reduce the cognitive burden on prescribers using PGx testing. Five studies [33,34,39,48,49] reported how an inability to order PGx testing through usual IT workflows presented a barrier to prescribers ordering behaviours. In the latter implementation stages, uploading genotyping reports on the electronic medical record in a searchable format, enabled prescribers to interpret PGx results [34].

IT systems interoperability represented a major barrier to pharmacists within PGx testing roles. A feasibility study investigating the implementation of a pharmacist-led PGx testing service to community-based medical centres reported how pharmacists directly sent PGx results to the prescriber via an online server and the pharmacy record. Whilst this method was feasible for this setting, the inability of pharmacists to access a central electronic medical record impacted the pharmacist recommendations [37]. Furthermore, modelling this IT structure in other settings may be challenging. For example, a descriptive case study describing PGx implementation in a large academic centre reported holding PGx data on multiple IT systems led to poor trackability of lifetime genetic results [29]. A survey of patients undergoing pharmacogenomic testing through a pharmacist-led pharmacogenomic clinic showed patients preferred for test results to be incorporated in the medical record so other medical providers had access, facilitating PGx-guided decision making [51].

Designing IT PGx workflows that are intuitive to end users is also important. One US study investigating the implementation of PGx testing in a health system serving both primary and secondary care reported how IT workflows integrating PGx were co-designed by pharmacists, physicians, and nurses [28].

#### 3.3.2. Effort

The cognitive, physical, and emotional effort to undertake behaviours necessary for implementation was a major theme of the studies included. Effort affected most of the behaviours across the implementation stages and was reported in more than half of the papers included (59%, n = 16/27).

Barriers and enablers affecting cognitive effort were most likely to be mapped to the ‘Memory, attention and decision making’ TDF domain. Electronic prompts in the form of PGx clinical decision support tools enabled prescribers to order PGx testing for patients [34] as well as enabling the interpretation and application of PGx test results by prescribers and pharmacists [29,33,34,37]. Health professionals pre-existing procedural competence meant that behaviours such as prescribers and pharmacists collecting DNA samples and sending them to a laboratory for testing were of low cognitive effort and easily implemented [32,44,50].

Physical effort emerged as a barrier to patients consenting to PGx testing. A survey of patients who had taken part in a PGx testing programme in the US reported nearly half of the participants (42%) (n = 869) were unwilling to incur out-of-pocket costs for PGx testing [52]. This was also found in a service evaluation of a US hospital implementing PGx testing, where reimbursement of testing was a significant barrier to patient engagement [42]. In addition to cost, DNA collection methods also represented a physical effort barrier to patient behaviours related to PGx implementation [34,44]. A feasibility study exploring a community pharmacy implementation model in the Netherlands, found saliva sampling to be challenging for certain groups of patients due to comorbidities or concurrent medicines [44]. This could be overcome by restructuring the environment and providing additional resources for example offering multiple DNA collection methods of blood, and saliva [34,49].

Electronic prompts were reported to reduce the cognitive effort of prescribers ordering, interpreting, and applying PGx results. The introduction of these alerts within clinical workflows was sometimes perceived negatively, and doctors reported alert fatigue if electronic prompts appeared indiscriminately for every patient [31,35]. Prescribers in primary care perceived PGx testing as complex and too specialised to use in their own practice, and this was exacerbated by unfamiliar nomenclature used in reporting results [31]. Emotional effort was, therefore, a complex theme that covered multiple TDF domains: ‘Social/professional role and identity’; ‘Emotion’, and ‘Optimism’.

#### 3.3.3. Rewards

Rewards as a theme described factors which were perceived by prescribers, pharmacists, or patients as a positive outcome to PGx testing. ‘Optimism’ and ‘Belief about consequences’ were the two most frequently mapped TDF domains for determinants under this theme. Patients′ reported optimism for a pharmacist PGx delivery model enabled patient consent behaviours within these implementation models [30,51]. Optimism on the part of the patient that the PGx testing would help their medical management enabled patient consenting behaviours [39,43,47,48] whereas pessimism on the part of the patient about the utility of PGx testing prevented these behaviours [45]. Optimism also impacted behaviours of prescribers related to PGx implementation. The perceived clinical utility or value for money of the test impacted whether a prescriber would order or apply a PGx test. Primary care physicians interviewed as part of a feasibility study investing PGx implementation in a rural US setting found the cost of PGx testing was a barrier to initiating testing suggesting a poorer perceived cost–benefit ratio [39]. In contrast, a survey of prescribers at a tertiary centre in the US reported favourable attitudes to the perceived clinical utility of testing enabling PGx testing applications [38].

Belief about consequences emerged as both a barrier and enabler to prescriber behaviours related to PGx implementation. This determinant centred on the prescriber’s perceptions about the clinical utility of PGx testing and was augmented by the clinical relevance of the drug–gene pairs implemented locally through the frequency or severity of drug–gene interactions encountered [26,35].

Turnaround time between testing and receiving results was also reported as a barrier to prescriber ordering behaviours [33]. This was overcome in several US implementation sites through environmental restructuring to enable a pre-emptive PGx testing approach [26,27].

#### 3.3.4. Unknown Territory

The novelty of PGx testing affected all stages of the implementation cycle but manifested as primarily a barrier at the initial stage of prescriber and pharmacist ordering behaviour. General knowledge of PGx and identifying patients for testing were reported as barriers to prescribers and pharmacist ordering behaviours [26,31,39,40]. In addition, a survey of prescribers and pharmacists at a tertiary centre with an established pharmacogenomic testing program, stated the greatest barrier to using PGx testing was an absence of established or clear guidelines for interpreting and applying results [40].

The lack of general PGx experience by prescribers affected prescriber confidence in using PGx. Prescribers were reported to hold negative beliefs about their capability to use PGx, consequently affecting their behaviours involving ordering, interpreting, and applying PGx information in clinical care [26,31]. Prescribers who had prior exposure to PGx information were reported to be more informed and confident in undertaking behaviours relating to PGx simply through experience [38,46].

In a backdrop of legal uncertainty two PGx implementation sites in the US, adopted a team approach to PGx interpretation, with a specific consult group managing and taking responsibility for liability associated with incidental findings [34,36]. It was not reported in these studies whether the drive for liability protections came from the prescribers themselves or the organisation.

## 4. Discussion

To the best of our knowledge, this is the first systematic review in the context of pharmacogenomic testing to focus on barriers and enablers to the behaviours of prescribers, pharmacists, and patients, relating to implementation in primary and secondary care and subsequently map them to the theoretical domain framework.

In line with previous research, information technology was identified as both a barrier to and an enabler of implementation [53]. A recent structured scoping review of pharmacogenetic testing programs using the CFIR found that IT solutions are currently unable to support pharmacogenomic-guided prescribing at the interface between primary and secondary care. This was a persistent problem to wider adoption and implementation [54]. At the individual level, we found that clinical decision support systems (CDSS) when linked to the electronic health record (EHR) in particular, enabled initiation and application of PGx testing through the mechanism of environmental restructuring and prompting prescriber PGx-related behaviours. The importance of well-designed CDSS alerts has been well-documented with the Dutch Pharmacogenetics Working Group describing one the earliest examples of implementing CDSS alerts in a national electronic prescribing and medicines surveillance system [55]. Since then, the Clinical Pharmacogenetics Implementation Consortium’s (CPIC) Informatics Working Group has provided best practice suggestions for integrating pharmacogenomics CDSS for clinical delivery [56]. Whilst our findings are not necessarily novel, given that IT interoperability [57] and CDSS design [58] have been the subject of extensive research, our linking to behavioural change theory may provide better direction for the future design and evaluation of effective CDSS which incorporate behavioural change techniques [59].

Prescriber and pharmacist views on the clinical utility and cost-effectiveness modulated their perceptions of the rewards of PGx testing. This finding corroborated a recent systematic review exploring barriers and enablers of PGx testing in primary care which also found the domain ’belief about consequences’ was an important driver for primary care physicians′ adoption of PGx testing [60]. Our findings show this domain influences physicians in secondary care and pharmacists in both settings. Our findings are also strengthened by excluding studies which consider attitudes towards PGx testing from a theoretical perspective.

Healthcare systems represent complex environments comprising multiple interacting components that are evolving dynamically and are interdependent [61]. PGx clinical implementation strategies often demanded new models of care thus adding to the complexity and effort required by people within the system to adapt and sustain PGx testing. The more a new intervention such as PGx demands different processes within an organisation, the more effort is required of the existing workforce, and the less likely it is to be taken up and sustained [62]. These new models of care were dominated by pharmacist-led models of implementation. The large emotional effort on the part of physicians to implement PGx testing arising from unfamiliarity and complex processes, led to them feeling that it did not align with their professional role and identity. This misalignment may be the driver for the more prominent pharmacist roles reported [63] which has been facilitated by advocacy for this role from pharmacy professional bodies in the US [64]. In contrast, medical professional statements have been confusing. For example, while Clopidogrel FDA labelling recommends alternative therapy in patients identified as CY2C19 poor metabolisers [65]. The American College of Cardiology Foundation, American Heart Association, and the Society for Cardiovascular Angiography and Interventions guidelines recommend against routine PGx testing in all percutaneous intervention patients [66]. Inconsistencies in messaging may negatively influence prescriber attitudes to PGx testing as professional associations play a key role in shaping the professional role and identity of their members [67].

Panel pre-emptive PGx testing is often cited as the most suitable model for implementing routine PGx testing in clinical care [68]. The reasons underpinning this predominantly centre around cost-effectiveness, with the aggregate effect of PGx testing on health outcomes being more favourable in a pre-emptive multiple drug–gene testing scenarios over a patient′s lifetime than through a single drug–gene reactive testing scenario [69]. Several studies demonstrate how common drug–gene interactions are in a wide range of populations [70,71,72,73]. However, each of these studies whilst describing a panel pre-emptive pharmacogenomic test uses different drug–gene pairs and guidelines for hypothetical implementation. This reflects wider discord over which genetic variants comprise a panel PGx testing approach that maximises clinical impact and is equitable and fair. There is yet no consensus over what a standardised panel is, however, there have been a few recent papers that have suggested prototypes for implementation in different contexts [5,74]. In this way, despite over two decades of research, implementation efforts of PGx testing in clinical care are challenged by the evolving, dynamic definitions of what PGx testing is and its constituent parts. As a result, the belief among health professionals that PGx testing is a novelty remains since familiarity with one form of testing may not translate to easing the use of another. This is perhaps reflected by the absence of endorsement for PGx testing by professional organisations [75] and complicated by the activities of private companies in this space.

### 4.1. Implications for Future Research

The majority of articles included in this review focused on the barriers and enablers to the prescriber and pharmacist behaviours related to implementation. The barriers and enablers were predominately described through author interpretations recounted in narrative descriptions of implementation rather than the primary data derived through traditional qualitative research methods. None of the articles used implementation frameworks or theory which introduces a degree of uncertainty to our findings.

Future research exploring determinants of the behaviours of physicians, pharmacists, and patients in real-world PGx implementation settings would be strengthened through the use of rich qualitative research methods and a theoretical lens. This would support the understanding of context-specific barriers and enablers (for example, in primary versus secondary care) and develop evidence-based, theory-informed interventions for the most appropriate implementation configuration.

### 4.2. Limitations

Limitations to this review relate to both the individual articles and review methodology. As discussed previously, almost all articles included described the authors’ interpretation of barriers and enablers versus first-person accounts of prescribers, pharmacists, and patients. To capture the full breadth of available real-world data, articles with a high risk of bias such as descriptive case studies were included. However, their findings were often reflected in studies with a low risk of bias. Only articles published in the English language were included due to resource constraints. The review was strengthened by adhering to PRISMA guidelines and use of an established theoretical framework to map and synthesise findings.

## 5. Conclusions

Multi-drug pharmacogenomic testing represents a complex intervention. Framing implementation through a behavioural science lens provides insight into the key determinants driving the behaviours of prescribers, pharmacists, and patients related to PGx testing. Memory, attention, and decision making, as well as beliefs about consequences and environmental context and resources, underpinned the main barriers to behaviours related to PGx testing implementation. Theory-based implementation interventions targeting these domains may facilitate the progression of efforts for widespread PGx adoption and sustainability.

## Figures and Tables

**Figure 1 jpm-12-01821-f001:**
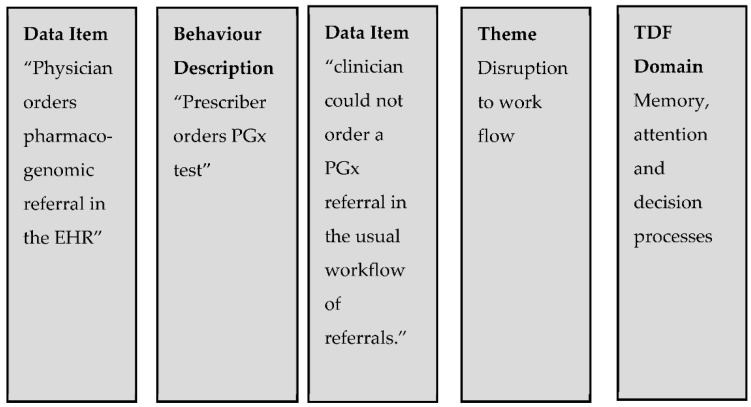
Data synthesis example.

**Figure 2 jpm-12-01821-f002:**
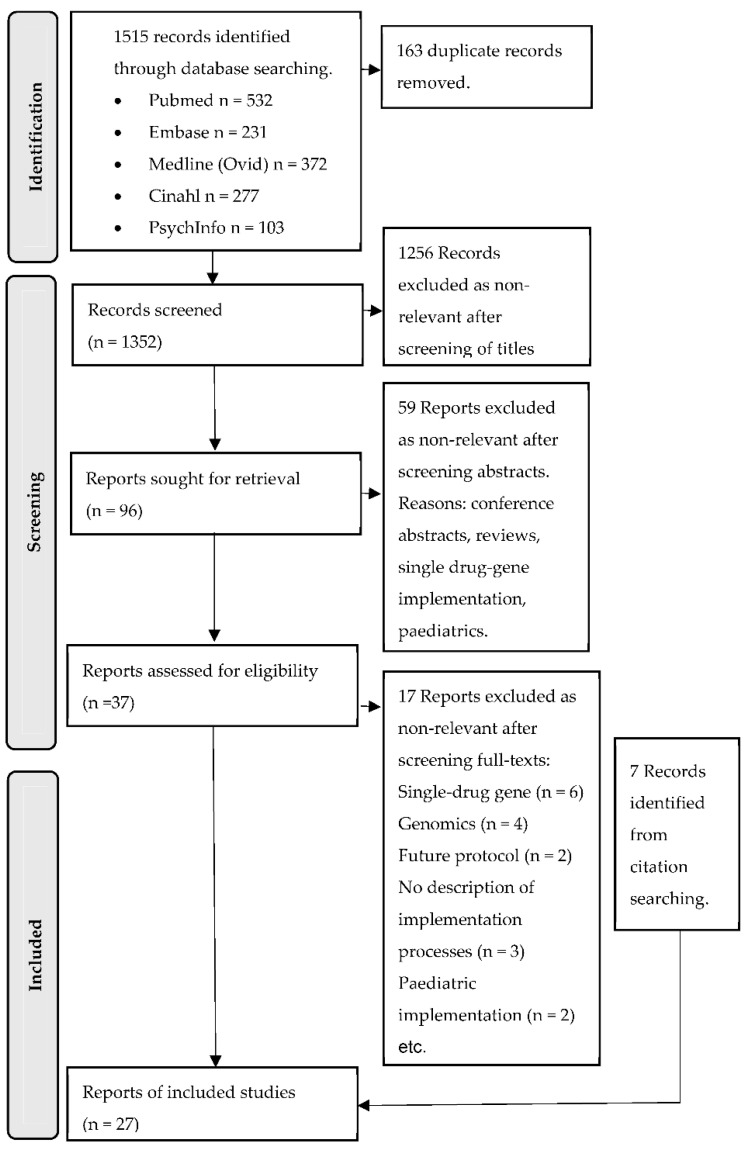
PRISMA flow diagram. PRISMA = Preferred Reporting Items for Systematic Reviews and Meta-Analysis.

**Figure 3 jpm-12-01821-f003:**
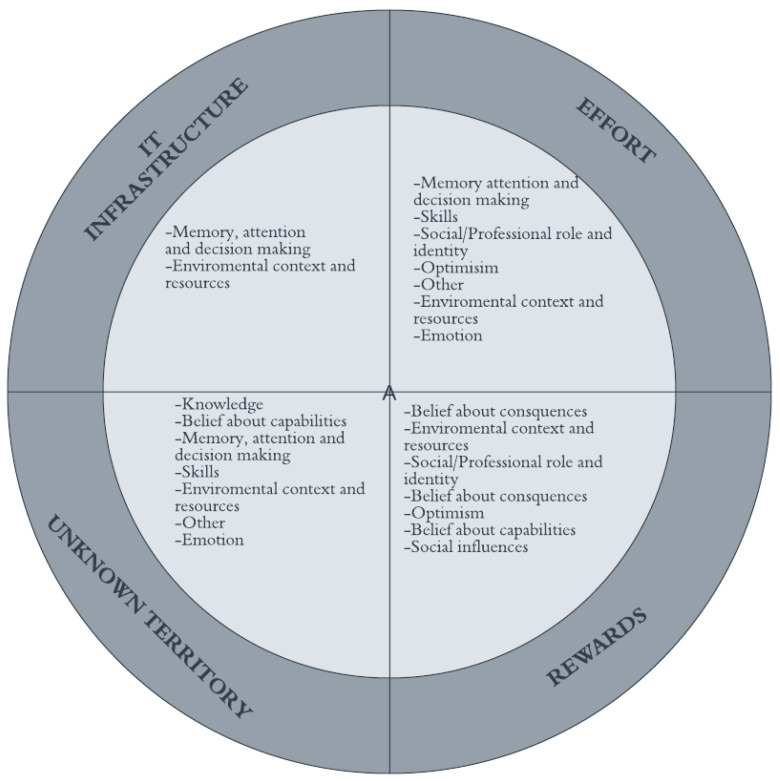
Four overarching themes emerged from the barriers and enablers extracted from the included papers. These themes (outer ring) are linked to TDF domains (inner four quadrants).

**Table 1 jpm-12-01821-t001:** TDF domains in context.

TDF Domain	TDF Domain Definition [13]	Definition in Context
**Knowledge**	An awareness of the existence of something.	Awareness of pharmacogenomics by prescribers, pharmacists, and patients.
**Skills**	An ability or proficiency acquired through practice.	The ability or proficiency prescribers, pharmacists, or patients have acquired to use pharmacogenomics through practice.
**Social/Professional Role and Identity**	A coherent set of behaviours and displayed personal qualities of an individual in a social or work setting.	The perceived professional role and personal identity of prescribers, pharmacists, and patients in relation to using pharmacogenomics.
**Belief about capabilities**	Acceptance of the truth, reality, or validity about an ability, talent, or facility that a person can put to constructive use.	Perception of prescribers, pharmacists, and patients about their own capability to use pharmacogenomics.
**Optimism**	The confidence that things will happen for the best or that desired goals will be attained.	The confidence, or otherwise, of prescribers, pharmacists, or patients around the use of pharmacogenomics in their practice.
**Belief about consequences**	Acceptance of the truth, reality, or validity about outcomes of a behaviour in a given situation.	Belief of prescribers, pharmacists, or patients about the value of using pharmacogenomics in their practice.
**Reinforcement**	Increasing the probability of a response by arranging a dependent relationship, or contingency between the response and a given stimulus.	Incentives, rewards, sanctions, and reinforcement from any level, including patient feedback, clinician perspectives, funding, and external views that facilitate the use of pharmacogenomics in practice.
**Intentions**	A conscious decision to perform a behaviour or a resolve to act in a certain way.	Intentions of prescribers, pharmacists, and patients to consider using pharmacogenomics in their practice.
**Goals**	Mental representations of outcomes or end states that an individual wants to achieve.	Perceptions by prescribers, pharmacists, and patients that pharmacogenomics can be potentially used in their practice.
**Memory, Attention and Decision Processes**	The ability to retain information, focus selectively on aspects of the environment and choose between two or more alternatives.	The ability for prescribers, pharmacists, and patients to remember to consider using pharmacogenomics.
**Environmental Context and Resources**	Any circumstances of a person’s situation or environment that discourages or encourages the development of skills and abilities, independence, social competence, and adaptive behaviour.	Any circumstance of the organisations situation or environment that discourages or encourages the ability of prescribers, pharmacists, or patients to use pharmacogenomics in practice including independence, social competence, and adaptive behaviour.
**Social Influences**	Those interpersonal processes that can cause individuals to change their thoughts, feelings, or behaviours.	Interpersonal interactions within and outside the organisation that can influence the thoughts, feelings, or behaviours of prescribers, pharmacists, or patients in relation to the use of pharmacogenomics.
**Emotions**	A complex reaction pattern, involving experimental, behavioural, and physiological elements, by which the individual attempts to deal with a personally significant matter or event.	Feelings by prescribers, pharmacists, or patients related to the use of pharmacogenomics in their practice.
**Behavioural Regulation**	Anything aimed at managing or changing the objectives of the observed or measured actions.	Anything prescribers, pharmacists, or patients have proactively created to help make decisions about and make changes in using pharmacogenomics.

**Table 2 jpm-12-01821-t002:** Summary of included studies.

**Study (Year) Country**	**Objective**	**Study Design**	**Study Setting**	**Methods Used**	**Actor**
Bain et al. (2018) USA	To determine the feasibility of implementing a pharmacist-led pharmacogenomics (PGx) service.	Feasibility Study.	Primary care. (community pharmacy).	Document analysis.	Prescriber.
Formea et al. (2015) USA	To describe experiences of implementing pharmacogenomics education in a large, academic healthcare system.	Descriptive case study.	Primary care.	Senior stakeholder observation.	Prescriber.
Bielinski et al. (2017) USA	To assess patient experiences and understanding of pharmacogenomics and pharmacogenomics educational materials.	Service evaluation.	Secondary care.	Survey.	Patient.
Dawes et al. (2017) Canada	To assess the ability to obtain and genotype saliva samples and determine levels of use of a pharmacogenomic decision support tool.	Prospective cohort study.	Primary care.	Document analysis.	Prescriber, Pharmacist.
O′Donnell et al. (2012) USA	To describe an institutional pharmacogenomics-implementation project.	Descriptive case study.	Secondary care.	Senior stakeholder observation.	Prescriber.
Haga et al. (2015) USA	To assess the feasibility of a combined pharmacist-delivered medication therapy management (MTM) with pharmacogenetic (PGx) testing.	Feasibility study.	Primary care.	Document analysis, survey.	Prescriber, Pharmacist.
Borden et al. (2019) USA	To understand whether pharmacogenomic results are discussed between patient and provider and whether medication recall is impacted by pharmacogenomic testing.	Service evaluation.	Primary care.	Survey.	Prescriber.
**Study (Year) Country**	**Objective**	**Study Design**	**Study Setting**	**Methods Used**	**Actor**
Levy et al. (2014) USA	To describe the key requirements to ensure a successful and enduring PGx implementation within a large healthcare system.	Descriptive case study.	Secondary care.	Senior stakeholder observation.	Prescriber, Pharmacist.
Dunnenberger et al. (2016) USA	To describe the development and implementation of a multidisciplinary pharmacogenomics clinic within a community-based medical genetics program.	Descriptive case study.	Secondary care.	Senior stakeholder observation.	Prescriber, Pharmacist.
Swen et al. (2012) Netherlands	To investigate the feasibility of pharmacy-initiated pharmacogenetic screening in primary care.	Feasibility study.	Primary care.	Document analysis, survey.	Pharmacist.
Bielinski et al. (2014) USA	To report the design and implementation of a pre-emptive pharmacogenomics (PGx) testing programme.	Descriptive case study.	Primary care, Secondary care.	Survey.	Patient.
Eadon et al. (2016) USA	To describe the formation of a pharmacogenomics consultation service at a safety-net hospital, which predominantly serves low-income, uninsured, and vulnerable populations.	Descriptive case study.	Secondary care.	Document analysis, Senior stakeholder observation.	Prescriber.
Unertl et al. (2015) USA	To describe the knowledge and attitudes of clinicians participating in a large pharmacogenomics implementation program.	Process evaluation.	Primary care, Secondary care.	Interviews.	Prescriber.
St Sauver et al. (2016) USA	To summarise and describe early clinician experience with pharmacogenomics in the clinical setting.	Service evaluation.	Secondary care.	Survey.	Prescriber.
Rosenman et al. (2017) USA	To describe challenges and potential solutions based on a pharmacogenomic testing programme.	Descriptive case study.	Secondary care.	Senior stakeholder observation.	Prescriber, Patient.
**Study (Year) Country**	**Objective**	**Study Design**	**Study Setting**	**Methods Used**	**Actor**
Moeddeb et al. (2015) USA	To characterise the experiences and feasibility of offering pharmacogenetic (PGx) testing in a community pharmacy.	Feasibility study.	Primary care (community pharmacy).	Document analysis.	Pharmacist, Patient.
Dressler et al. (2019) USA	To assess the feasibility and perspectives of pharmacogenetic testing in rural, primary care physician practices.	Feasibility study.	Primary care.	Survey.	Prescriber, Patient.
Arwood et al. (2020) USA	To describe the development, workflow, and early implementation challenges associated with a pharmacist pharmacogenetic testing clinic.	Service evaluation.	Secondary care.	Document analysis, Senior stakeholder observation.	Prescriber, Pharmacist.
Bright et al. (2020) USA	To evaluate the implementation processes relating to a pharmacist pharmacogenetic testing consult service.	Service evaluation.	Secondary care.	Document analysis, Senior stakeholder observation.	Pharmacist.
Haga et al. (2021) USA	To assess pharmacist experiences with delivering pharmacogenetic testing in independent community pharmacies.	Process evaluation.	Primary care.	Survey, Document analysis, semi-structured interviews.	Pharmacist.
Lanting et al. (2020) Netherlands	To identify barriers and facilitators to the implementation of an outpatient pharmacogenetic screening service.	Process evaluation.	Secondary care.	Survey, interviews, focus group.	Pharmacist, Patient.
Liko et al. (2021) USA	To describe the implementation of a pharmacist-provided pharmacogenomic testing service at an academic medical centre.	Descriptive case study.	Secondary care.	Senior stakeholder observation.	Pharmacist.
Marrero et al.(2020) USA	To describe the transition from implementing single-gene testing to a pre-emptive panel-based pharmacogenetic testing service.	Descriptive case study.	Secondary care.	Senior stakeholder observation.	Prescriber, Pharmacist.
**Study (Year) Country**	**Objective**	**Study Design**	**Study Setting**	**Methods Used**	**Actor**
Tuteja et al. (2021) USA	To evaluate the approaches taken by early adopters to implement a clinical pharmacogenetic testing service.	Service evaluation.	Primary care, Secondary care.	Survey.	Prescriber, Pharmacist.
Van der Wouden et al. (2020) Netherlands	To identify pharmacists’ perceived barriers and enablers facilitating the implementation of pharmacist-initiated pharmacogenetic testing in primary care.	Service evaluation.	Primary care.	Interview, Survey.	Pharmacist.
Ho et al. (2021) USA	To characterise clinician perceptions, practices, preferences and barriers to integrating pharmacogenomics in a single pharmacogenomic clinic.	Service evaluation.	Secondary care.	Survey.	Prescriber.
Martin et al. (2022) USA	To assess the perspectives and experiences of patients participating in a pharmacist-led PGx service.	Service evaluation.	Tertiary care.	Semi-structured interviews.	Patient, Pharmacist.

**Table 3 jpm-12-01821-t003:** Barriers and enablers reported for each behaviour.

**Implementation Step**	**Description of Behaviour**	**Theme**	**TDF Domain**	**Perspective**	**Reported Barrier**	**Reported Enabler**
Ordering test	Prescriber orders PGx test	IT Infrastructure	Memory, attention, and decision making	Prescriber	- Disruption to workflow [33]	- No data available
					- Logistics/ease of use [39]	- No data available
		Effort	Memory, attention, and decision making	Prescriber	- Perceived additional workload of test [33,36]	- No data available
					- Paperwork [47]	- No data available
			Skills	Prescriber	- Unclear procedures [50]	- Previous exposure to PGx [37,49]
			Social/Professional role and identity	Prescriber	- Language of result reporting [31]	- No data available
			Optimism	Prescriber	- Perceived complexity of PGx [31]	- No data available
			Other	Prescriber	- Content and form of training [31]	- No data available
					- Low clinician engagement [35]	- No data available
	Pharmacist orders PGx test	Rewards	Belief about consequences	Pharmacist	- No data available	- Pharmacist’s perceived value of testing [40,51]
		Unknown territory	Knowledge	Pharmacist	- Awareness of availability of testing [40]	- No data available
**Implementation Step**	**Description of** **Behaviour**	**Theme**	**TDF Domain**	**Perspective**	**Reported Barrier**	**Reported Enabler**
Ordering test	Prescriber orders PGx test	Rewards	Belief about consequences	Prescriber	- No data available	- Prescriber perceived value of testing [37,39]
			Environmental context and resources	Prescriber	- Demand/supply for service [33]	- No data available
				Prescriber	- Disruption to workflow due to time delay for results [52]	- No data available
		Unknown territory	Belief about capabilities	Prescriber	- Perceived confidence to order test [31]	- No data available
			Memory, attention, and decision making	Prescriber	- Liability of incidental findings [36]	- Ability to recognise drug–gene pairs [34,52]
			Skills	Prescriber	- Prescriber knowledge of who to test [39,40]	- No data available
			Environmental context and resources	Prescriber	- Reimbursement [40,50]	- No data available
			Knowledge	Prescriber	- Knowledge gap when to order test [37]	- No data available
				Prescriber	- Awareness of availability of testing [40]	- No data available
			Other	Prescriber	- Availability of guidelines [40]	- No data available
	Pharmacist orders PGx test	Effort	Environmental context and resources	Pharmacist	- Reimbursement [40,47]	- No data available
		Rewards	Social/Professional role and identity	Pharmacist	- No data available	- Pharmacist expert knowledge [50,51]
**Implementation Step**	**Description of** **Behaviour**	**Theme**	**TDF Domain**	**Perspective**	**Reported Barrier**	**Reported Enabler**
Facilitating test	HCP collects pts DNA sample	Effort	Skills	Pharmacist	- No data available	- Procedural competence [32]
			Other	Patient	- Physical challenge providing test specimens [44]	- No data available
	Patient gives consent to PGx test	Effort	Environmental context and resources	Patient	- Cost [42,52]	- No data available
			Emotion	Patient	- No data available	- Patient acceptability of DNA collection method [34]
		Rewards	Belief about consequences	Patient	- Perceived utility of test [36,45]	- No data available
			Optimism	Patient	- Pessimism about test ultilty [45]	- Perception of the test will be useful [39,47,48] - Confidence in pharmacist knowledge [30,51]
		Unknown territory	Emotion	Patient	- Perceived risk of discrimination [33]	- No data available
					- Concerns about data privacy [36]	- No data available
					- Perceived implications for family members [33]	- No data available
	Pharmacist shares report with prescriber	IT Infrastructure	Environmental context and resources	Prescriber, Pharmacist	- Information technology interoperability [37]	- No data available
**Implementation Step**	**Description of** **Behaviour**	**Theme**	**TDF Domain**	**Perspective**	**Reported Barrier**	**Reported Enabler**
Facilitating the test	HCP counsel′s patient on PGx result	Effort	Environmental context and resources	Prescriber, Pharmacist	- Prescriber and pharmacist access to central prescribing system [37,51]	- No data available
		Rewards	Environmental context and resources	Patient	- No data available	- Patient access to report results [43]
		Unknown territory	Skills	Prescriber	- No data available	- Prescriber experience with PGx [38]
Interpretating the test	Pharmacist interprets PGx results	Effort	Social/Professional role and identity	Prescriber, Pharmacist	- No data available	- Pharmacist expert knowledge [33,42,46]
	Prescriber interprets PGx result	Effort	Memory, attention, and decision making	Prescriber	- No data available	- Electronic workflow alert for drug–gene pairs [34]
			Emotion	Prescriber	- Negative perception of CDS [35]	- No data available
			Social/Professional role and identity	Prescriber	- No data available	- Pharmacist expert knowledge [50]
		IT Infrastructure	Memory, attention, and decision making	Prescriber	- No data available	- Location of results [34]
**Implementation Step**	**Description of** **Behaviour**	**Theme**	**TDF Domain**	**Perspective**	**Reported Barrier**	**Reported Enabler**
Application of the test	Prescriber applies PGx result	Effort	Memory, attention and decision making	Prescriber	- Location of results [43]	- No data available
		IT Infrastructure	Environmental context and resources	Prescriber	- No data available	- CDS alert at point of prescribing [29,48,49]
		Rewards	Belief about capabilities	Prescriber	- Prescriber perceived lack of capability to apply results [26]	- No data available
			Belief about consequences	Prescriber	- Perceived severity of drug–gene interaction [35]	- Perceived utility of PGx testing [38]
			Social influences	Prescriber	- Perceived utility of drug–gene pairs [26]	- No data available
		Unknown territory	Environmental context and resources	Prescriber	- Prescriber liability [34]	- No data available
			Knowledge	Prescriber	- Knowledge gap on how to apply PGx results [26,37]	- No data available

## Data Availability

The datasets generated and analysed during the current study are available from the corresponding author upon reasonable request.

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
