# Peer review of "A Theory-Informed Systematic Review of Barriers and Enablers to Implementing Multi-Drug Pharmacogenomic Testing"

_jpm, 2022, doi:10.3390/jpm12111821_

Round 1
Reviewer 1 Report
This article describes a systematic review of multi-gene pharmacogenomic implementations where the Theoretical Domains Framework (TDF) was applied to organize and report barriers or enablers to pharmacogenomic testing. The article adequately describes the search approach used to identify relevant articles and the data synthesis process. The authors describe their process in elucidating 11 activities related to 4 implementation stages, across which four themes were identified. This work provides a novel and more systematic approach to identifying facilitators and barriers to implementing multi-gene PGx testing.
Some suggestions and questions for the authors:
1. On line 26 there is a typo: “globally” should be” global.
2. In the first paragraph describing the methods, the end of line 64-line 71 seem to be instructions for submitting the paper and not actual methods of the paper. Is this supposed to be there?
3. Why was single-gene testing excluded? Would be relevant to include that explanation, especially in the discussion section that mentions single-gene testing. This would help to provide some context that facilitators and barriers for single-gene testing could be different.
4. Consider adding clinical specialty targeted for implementation to table 2. This could provide some additional context for the facilitators and barriers.
5. On line 295 there is a typo: it should read “Designing IT PGx workflows”
6. The last paragraph of results section 3.3.2 Effort is somewhat confusing in the way it is written. The first sentence seems to be missing information or should be connected to the next sentence. Also, what is meant by “Prescribers in primary care, perceived PGx testing as complex and specialist to use it in their own practice”. Should this read “too specialized”?
7. It would be helpful to have some sort of visual representation of the frequency with which each TDF domain was referenced within each theme or the frequency as a barrier vs. facilitator. Table 3 does provide the information for what was observed, but some sort of visual summary to show the frequency would be helpful for readers.
8. The introduction states that part of the purpose of this paper is to identify the most appropriate configuration for PGx testing implementation. However, this does not seem to actually be discussed in the paper.
9. A study was recently published that characterized pgx implementations utilizing CFIR. This paper also discusses facilitators and barriers. It would be prudent to discuss any major similarities or differences between the two approaches. (PMID 36059837).
Reviewer 2 Report
This systematic review analyzes the barriers and enablers from a physician, pharmacist, and patient perspective of multi-gene pharmacogenomic testing with an implementation science approach. This adds to the rising need for an evidence-based implementation science understanding to clinical pharmacogenomic implementation. Although the author uses the Theoretical Domain Framework (TDF) to evaluate the themes of the barriers and enablers from diverse perspective, further discussion on how this framework was used in similar non-PGx healthcare implementations (non-PGx interventions) or how it compares to other implementation science frameworks/models or similar studies such as the consolidated framework of implementation research (CFIR) reported by Sony Tuteja et al. (PMID 34562070) or Sadaf Qureshi et al. (PMID 34911350) systematic review of pharmacogenomic testing in primary care. Additional knowledge gap that should be consider is why this review excludes single-gene PGx and pediatrics implementation. Please see the word document attached with specific comments on omitted/duplicated citations, suggestions to improve understandability of figures/tables, and missing information in figures/tables.
